# Recent Advances in Strategies for Activation and Discovery/Characterization of Cryptic Biosynthetic Gene Clusters in *Streptomyces*

**DOI:** 10.3390/microorganisms8040616

**Published:** 2020-04-24

**Authors:** Chung Thanh Nguyen, Dipesh Dhakal, Van Thuy Thi Pham, Hue Thi Nguyen, Jae-Kyung Sohng

**Affiliations:** 1Department of Life Science and Biochemical Engineering, Sun Moon University, 70 Sunmoon-ro 221, Tangjeong-myeon, Asan-si, Chungnam 31460, Korea; chungnguyenbio@gmail.com (C.T.N.); medipesh@gmail.com (D.D.); phamtthuyvan@gmail.com (V.T.T.P.); huenguyencute@gmail.com (H.T.N.); 2Department of Pharmaceutical Engineering and Biotechnology, Sun Moon University, 70 Sunmoon-ro 221, Tangjeong-myeon, Asan-si, Chungnam 31460, Korea

**Keywords:** *Streptomyces*, natural product, biosynthetic gene cluster, cryptic, activation

## Abstract

*Streptomyces* spp. are prolific sources of valuable natural products (NPs) that are of great interest in pharmaceutical industries such as antibiotics, anticancer chemotherapeutics, immunosuppressants, etc. Approximately two-thirds of all known antibiotics are produced by actinomycetes, most predominantly by *Streptomyces*. Nevertheless, in recent years, the chances of the discovery of novel and bioactive compounds from *Streptomyces* have significantly declined. The major hindrance for obtaining such bioactive compounds from *Streptomyces* is that most of the compounds are not produced in significant titers, or the biosynthetic gene clusters (BGCs) are cryptic. The rapid development of genome sequencing has provided access to a tremendous number of NP-BGCs embedded in the microbial genomes. In addition, the studies of metabolomics provide a portfolio of entire metabolites produced from the strain of interest. Therefore, through the integrated approaches of different-omics techniques, the connection between gene expression and metabolism can be established. Hence, in this review we summarized recent advancements in strategies for activating cryptic BGCs in *Streptomyces* by utilizing diverse state-of-the-art techniques.

## 1. Introduction

Natural products (NPs) derived from microbes generally possess diverse ecological and environmental impacts such as altering phenotypes, fitness, and community composition of microbes in the context of the environmental factors or ecological settings [1]. Nevertheless, in the context of human welfare, these NPs have been proven to be wonder molecules with diverse biological activities, such as antibacterials, anticancer agents, antihelminths, antidiabetics, anticholesterols, immunosuppressants, etc. Thus, they are important lead targets in the field of drug discovery and development [2,3]. Approximately 80% of anticancer agents and roughly 50% of all Food and Drug Administration-approved drugs are derived from such NPs [4]. Among all microorganisms, the actinomycetes are a prolific source of such valuable compounds that are of great interest to medicine, agriculture, and industry [5,6]. Approximately two-thirds of all known antibiotics are able to be produced by actinomycetes, most predominantly by *Streptomyces* [7]. Similarly, various species of *Streptomyces,* have been characterized as major producers of anticancer drugs [8]. In addition, *Streptomyces* are capable of producing effective compounds with diverse bioactivities as antihelminths, anti-oxidants, immunosuppressants, herbicides, and insecticides [9,10].

*Streptomyces* is a genus of Gram-positive and aerobic bacteria with high G+C content. Their taxonomic classification categorizes them within phylum: *Actinobacteria*, class: *Actinomycetes,* order: *Actinomycetales* and family: *Actinomycetaceae.* The genus *Streptomyces* was proposed by Waksman and Henrici [11]. Waskman and colleagues were pioneers for isolating bioactive compounds from various *Streptomyces*, such as actinomycin from a culture of *Streptomyces antibioticus* [12], streptothricin from *Streptomyces lavendulae* [13], and streptomycin from *Streptomyces griseus* [14]. These innovations led to the discovery, isolation, and evaluation of the bioactivity of diverse molecules from various *Streptomyces.*

Recent research indicated that the rate of new NP discovery has been maintained despite a drop in the number of compounds making it through regulatory approval pipelines [15]. However, the emergence of antibiotic-resistant pathogens or drug-resistant disease conditions and most of the NPs are not produced in significant titers or the biosynthetic gene clusters (BGCs) are cryptic, or the producer strains are not genetically tractable, which pose the greatest challenge for NPs based drug discovery [16,17,18]. In such cases production can be achieved in other production platforms as genetically tractable alternative hosts or suitable heterologous hosts [17,18,19].

The rapid development of genome sequencing technology has provided access to enormous numbers of BGCs in the microbial genomes. In addition, the advances in the studies of metabolism and metabolite profiling can also illustrate the entire network of metabolism and regulatory system contributing to biogenesis of the particular metabolite [19]. Thus, these multi-omics techniques have established a proper connection between genomic information, gene expression levels, and metabolism [20]. Thus, this knowledge provides new opportunities for discovering a larger diversity than those isolated to date [20,21]. All this progress has resulted in the development of a powerful method for NPs discovery termed “genome mining”. Unlike the traditional bioactivity directed way of isolating NPs, in genome mining, bioinformatics analysis of the sequenced microbial genomes can predict the BGC for NPs, which may be usually silent in the ordinary culture conditions, and are prospective targets to be activated [22,23]. Such genome mining approaches have been successfully utilized for obtaining isolatable quantities of compounds usually produced in low titers. Fundamentally, the genome mining-based NP characterization is mediated by two major strategies. The first approach can be tuning the production by altering physiological conditions, the use of elicitors/chemicals, and modulating the regulatory system in the native host. The second approach can be production in the rationally engineered heterologous host [18]. However, the heterologous expression suffers from various disadvantages, such as the limited efficiencies in the cloning of large BGCs; unavailability of a suitable expression platform; and incompatibility of the biosynthetic elements, such as biosynthetic precursors, regulations, and cofactors. These constraints provide strong support for the quest for mining NPs from native producers, including genetic and chemical induction-based approaches [18,24]. Moreover, it is evident that a single potent *Streptomyces* encodes more than 20–30 BGCs for diverse bioactive compounds among them most of them are cryptic, but it is not feasible to transfer all BGCs to heterologous host in the single setting. The holistic transfer of individual gene clusters in the single heterologous host or each gene cluster in separate heterologous hosts along with optimization of production parameters for target molecules is next to impossible effort. Therefore, the tuning or activation of the cryptic BGCs by the chemical modulation or the genetic intersection in the native host provides a tremendous opportunity for obtaining targeted, as well as untargeted NPs. Thus, this approach in most cases uncovers the unprecedented discovery of novel chemical structures with efficient biological activities [18]. In addition, the engineering/modulation steps may be less rigorous than the optimizations required in heterologous expressions, which requires rigorous steps of host selection and cloning, and in most cases, engineering of the producer host, optimizing the media components, and even the perturbation of regulation systems [24]. This review summarizes recent advancements in strategies for activation and discovery cryptic BGCs in *Streptomyces*, thus utilizing it as the most efficient native host for the production of promising bioactive molecules. 

## 2. Ribosome/RNA Polymerase Engineering Approach

“Ribosome engineering” was coined in 1996, when streptomycin was used to induce the mutation of *rpsL* (gene encoding ribosomal protein S12) in *Streptomyces lividans*, resulting in the activation of actinorhodin production. The point mutation occurred by changing of Lys-88 to Glu [25]. The nucleotide guanosine 5′-diphosphate 3′-diphosphate (ppGpp) plays an important role by responding to nutrient limitation, and initiating the biosynthesis of antibiotics, whereas ppGpp binds to RNA polymerase [26]. ppGpp is synthesized by ppGpp synthase (RelA), which is a subunit of the ribosome. The mutation of ribosomal protein S12 by streptomycin leads to an increase in the amount of ppGpp. As a result, the secondary metabolite production is enhanced or activated [27]. Challenging *Streptomyces* with streptomycin often introduces the mutation in ribosomal protein S12 in the spontaneous resistant strain. Additionally, it often introduces mutation in *rsmG* (encoding for 16S rRNA methyltransferase) in the resistant strain. The mutation in 16S rRNA methyltransferase can also enhance the S-adenosyl methionine (SAM) synthetase activity and the transcription level of secondary metabolism; which results in activating the cryptic pathway [28,29]. Similarly, rifampicin was used to activate the production of actinorhodin in *Streptomyces coelicolor* A3(2) and *S. lividans*. It was found that rifampicin induced mutation in *rpoB* (encoding the β-subunit of RNA polymerase). This mutation makes RNA polymerase mimic the function of ppGpp-RNA polymerase bound form, which leads to activation of the cryptic pathway [30,31,32]. Similarly, induction of combined drug-resistant mutations for different antibiotics such as streptomycin, gentamicin, and rifampin was used to continuously increase the production of antibiotics in a stepwise manner in *S. coelicolor.* The single, double, and triple mutants displayed a remarkable increase in the production of ActII-ORF4 (a pathway-specific regulatory protein) in the same hierarchical order as observed for the increase in the production titer of actinorhodin [33].

Further, the ribosome engineering strategy was used to activate antibiotic production from a large number of actinomycetes using streptomycin, rifampicin, or gentamicin [34]. In the test of different actinomycetes strains, two out of seven *Streptomyces* strains (efficiency 29%), and five out of 61 non-*Streptomyces* strains (efficiency 8%) were activated for secondary metabolites production. The technique was also applied to soil-isolated strains, where antibiotic production in 51 *Streptomyces* strains and 15 non-*Streptomyces* strains were activated successfully, among 119 *Streptomyces* strains and 234 non-*Streptomyces* strains, with the efficiency of 43% and 6%, respectively [34]. Similarly, rifampicin was used to induce the mutation in *rpoB* in *S. griseus*, *S. coelicolor*, *S. antibioticus*, *S. lavendulae*, *Streptomyces parvulus, Saccharopolyspora erythraea* (*Sac. erythraea*) and *Amycolatopsis orientalis* (*Amy. orientalis*) [34]. This led to enhanced production of streptomycin (*S. griseus*), actinorhodin (*S. coelicolor*), actinomycin (*S. antibioticus*), formycin (*S. lavendulae*), erythromycin (*Sac. erythraea*) and vancomycin (*Amy. orientalis*). Transcriptional level analysis of cryptic pathway in *S. griseus*, *S. coelicolor*, and *Sac. erythraea* by real-time PCR showed that the cryptic pathway was activated in the *rpoB* and *rsmG* strains, 3 to 70-fold increase in transcription [35]. Rifampicin was used for the activation of fredericamycin production in deep sea-derived *Streptomyces somaliensis* [36]. The efficiency of different antibiotics, such as rifampicin, streptomycin, paromomycin, erythromycin, and gentamicin were compared for enhancing antibiotic production and activating the silent gene in *Streptomyces diastatochromogenes*. The production of toyocamycin, tetramycin A, tetramycin P, and tetrin B were highest (4.1, 7.8, 5.1, and 12.9-fold, respectively) while using paromomycin to induce the mutation. Similarly, *rsmG* mutant strain of *S. diastatochromogenes* substantially increased the production of toyocamycin and also activated silent genes involved in the biosynthesis of secondary metabolites [37]. 

Similarly, the ribosome engineering strategy was employed by using several antibiotics together to generate a multiple drug-resistant strain. The *S. coelicolor* A3(2) resistant strain of seven or eight antibiotics (such as streptomycin, gentamycin, rifampicin, paromomycin, geneticin, fusidic acid, thiostrepton, and lincomycin) was generated, that produced 1.63 g/L of actinorhodin, that is 180-fold higher than wild-type strain [38]. The site-directed mutagenesis (SDM) strategy was also used to modify the *rpsL* in *S. lividans*. From a previous study, the K88E mutation of ribosomal protein S12 can be introduced by streptomycin, which leads to activate undecylprodigiosin production in *S. lividans*. Different individual point mutations including K88E mutant was generated by SDM and it was found that the L90K and R94G mutants can enhance undecylprodigiosin production by 2.9 and 1.9-fold higher than K88E mutant strains of *S. lividans,* respectively. However, the L90K or R94G mutant strains did not show an increasing level of resistance to paromomycin and streptomycin [39].

The ribosome engineering can be a simple, efficient, and time-saving strategy for activation of the silent gene cluster using common antibiotics. Moreover, the artificial induction of such mutations with the SDM method can be a very promising strategy for enhancing the production of compounds or activating the silent genes gene clusters, which could not be easily achieved by screening strains in antibiotic media. 

## 3. The OSMAC Approach Mediated by Co-Culture

More than a decade ago “One Strain Many Compounds” (OSMAC) was employed as a successful strategy for isolation and identification of up to 20 different new secondary metabolites from the *Streptomyces* [40]. OSMAC approaches cover different strategies, such as changing medium composition and cultivation status, or co-cultivation with other strain(s). The alteration of the media component has a prominent impact on obtaining new NPs from *Streptomyces* through the OSMAC approach. For example, the chemical investigation of the extract from the marine-derived strain *Streptomyces* sp. C34 grown on ISP2 (yeast malt extract agar) medium resulted in the isolation of four new ansamycin-type polyketides. However, only three compounds could be extracted from the modified ISP2 medium, which contained glycerol, rather than glucose [41]. *Streptomyces* sp. DSM 14386 produced five new compounds (**17–21**) in medium supplemented with 1.5% NaCl, while this strain produced two brominated congeners (**22,23**) in medium containing 1.5% NaBr [42]. Similarly, a novel antibacterial cyclodepsipeptide, named NC-1(**24**), was produced by a red soil-derived strain *Streptomyces* sp. FXJ1.172 when cultured in GYM medium supplemented with ferric ion [43].

Similarly, the changes in culture conditions have a prominent impact on tuning the production of novel compounds in the OSMAC approach. *Streptomyces* sp. CHQ-64 produced six new antifungal polyene-polyols reedsmycins A-F and two new cytotoxic hybrid isoprenoid alkaloids indotertine A and drimentine F in liquid medium under shaking condition, while this strain produced a new hybrid isoprenoid alkaloid drimentine I under static condition [44]. The culture of *Streptomyces* sp. HZP-2216E in solid medium as 2216E and GYM (glucose-yeast extract medium), and liquid medium as GMSS (Gause’s medium with sea salt) resulted in the isolation of two new compounds as 23-*O*-butyrylbafilomycin D, and streptoarylpyrazinole A, and a unique indolizinium alkaloid streptopertusacin A [45,46].

Recently, the OSMAC approach based on the co-culture method is a very popular approach of elicitation of cryptic biosynthetic pathways. The co-cultivation results in alteration of the environmental setting by incorporating the multi-dimensional interspecies interaction. This method not only involves physical interaction between the species but also involves cross-talks between their metabolic pathways or signaling cascades, thus, results in unprecedented activation of the biosynthesis of novel NPs from *Streptomyces.* The co-culture of *S. coelicolor* M145 with other actinomycete strains (*Amycolatopsis* sp. AA4, *Streptomyces* sp. E14, *Streptomyces* sp. SPB74, and *Streptomyces viridochromogenes* DSM 40736) resulted in the production of at least 12 different versions of a molecule called desferrioxamine [47]. Co-cultivation of *Streptomyces leeuwenhoekii* C34 and *Aspergillus fumigatus* MR2012 in ISP2 medium resulted in the yield of a new luteoride derivative, and a new pseurotin derivative, whereas none of these compounds could be detected in axenic culture. When *S. leeuwenhoekii* C58 was co-cultivated with *A. fumigatus* MR2012, a lasso peptide chaxapeptin was produced, which displayed significant inhibitory effect on human lung cancer cell line A549 [48,49].

## 4. Application of Rational Chemical Elicitors

The regulation of the NP production involves multiple regulatory cascades and internal networking, which has immense implications for the control of biosynthesis and the production of such NPs [50]. In secondary metabolism, there are two levels of regulations: (1) specific regulation for the pathway of biosynthesis, and (2) pleiotropic regulation for controlling the multiple pathways of biosynthesis. Pleiotropic regulators indirectly control the production of NPs, often located in clusters, or modulate the pathway-specific genes [51,52]. Elicitors are small molecules, acting as signal molecules, which can respond to many of the pleiotropic regulators [53]. The elicitors are very diverse. They can be antibiotics (chemical synthetic or biosynthetic compounds), inhibitors or activators of enzymes, or any compounds which can affect the metabolism of *Streptomyces* [52,54,55,56]. In 1967, the first elicitor, A factor, a γ-butyrolactone, was reported, which controls the biosynthesis pathway of streptomycin and spore formation [53,57]. Not only that, γ-butyrolactones regulate the production of a large class of antibiotics in different *Streptomyces*, including virginiamycins (*Streptomyces virginiae*) and showdomycin (*S. lavendulae*) (Figure 1B) [52,54]. PI factor, a 2,3-diamino-2,3-bis(hydroxymethyl)-1,4-butanediol, can activate the production of pimaricin (Figure 1B) in *Streptomyces natalensis*. Additionally, hydromethylfuran was shown to induce methylenomycin variants (Figure 1B) in *S. coelicolor* [55].

Thus, the application of new elicitation strategies can be an effective strategy for streamlining the activation of cryptic biosynthesis gene clusters. Several attempts of conventional screening methods using elicitor have been attempted to activate cryptic pathways. In 2012, a new approach of High-Throughput Elicitor Screens (HiTESs) of cryptic gene clusters was utilized to activate the silent biosynthesis pathways, whereas the *S. coelicolor* based model system was used. The production of the blue color compound (actinorhodin) and the red-colored compound (undecylprodigiosin) (Figure 1B) were used as the indicator compounds as they were easy to detect visually. Further 30,569 small molecules were used for screening the elicitors for activating the production of actinorhodin in *S. coelicolor*. Among them, 112 compounds affected, and of these, 19 compounds enhanced the production of actinorhodin. One of these compounds, ARC2, displayed pleiotropic regulations by enhancing the production of actinorhodin by 2 to 5-fold, and enhanced the production of germicidins (Figure 1B) to 3-fold, while decreasing the production of daptomycin-like calcium-dependent antibiotic by approximately 2-fold [52,58,59].

The major drawback of this method is that it is difficult to activate the particular silent gene cluster of interest. To solve this problem, a new strategy was developed by combining HiTESs with the reporter-guided genes. To test this idea, the cryptic BGC of malleilactone was activated in *Burkholderia mallei* by integrating the *lacZ* reporter gene to *malL*, a gene essential of the BGC of malleilactone. Screening elicitors by HiTESs based on LacZ activity from a library of 800 compounds, they successfully awakened the cryptic gene cluster to get different malleilactone analogs (Figure 1B) [60]. HiTESs combined with reported-guided strategy were also used in *Streptomyces,* for the first time for activating a *sur* cluster, the nonribosomal peptide synthetase (NRPS) gene cluster in *Streptomyces albus* and isolating different surugamide derivatives (Figure 1B). They used two reporter systems, XylE and eGFP, to apply HiTESs. They integrated the reported gene in the downstream of P*sur* promoter (the promoter driving the expression of the *sur* gene cluster). HiTESs were performed using 502 compounds as the elicitors, whereas two excellent elicitors, etoposide and ivermectin, could activate the *sur* cluster. The optimal concentrations for activating the *sur* cluster are at ≈23 μM (for etoposide), and ≈30 μM (for ivermectin), respectively [56]. Therefore, combining HiTESs with the reported-guided genes can be a very useful strategy to activate the silent biosynthesis gene clusters of interest. The principle of HiTESs combined reporter-guided screening approach is shown in Figure 1A. It can save time, in the case of many compounds that need to be induced in different strains, and new compounds are sought. However, the major disadvantage of this strategy is that when this approach is applied, it requires hundreds to thousands of chemicals for screening purposes, which may not available in every laboratory.

## 5. Transcription Factor Decoys Approach

Transcription factor decoys (TFDs) are short double-stranded DNA molecules that mimic the specific binding site in the promoter of a transcription factor. They can interfere with gene expression by aberrant binding between the transcription factor and the target promoter. When the copy number of the TFDs is large, the repressor or activator only binds with the TFDs. That leads to decreased binding of the repressor or activator with the promoter. The results are de-repression or de-activation of the target gene by the TFDs. The TFD strategy has been widely applied in mammalian cells and preclinical studies [61,62].

However, the TFD strategy was used for the first time to activate the cryptic BGCs in *Streptomycetes* [63]. As it is difficult to exactly predict the regulatory DNA fragments in the BGCs, DNA fragments of approximately 100 base pairs were cloned to both the upstream and downstream of open reading frames (ORFs) in the target BGC. These DNA fragments, the TFDs, were transferred into the native hosts. For proof of concept, they activated two silent BGCs of actinorhodin and undecylprodigiosin in *S. lividans* 66 by using the low-throughput TFD strategy. Five putative regulators in the BGC of actinorhodin and three putative regulators in the BGC of undecylprodigiosin were cloned as the TFDs, and each TFD was introduced into *S. lividans* individually. Multiple copies of the TFDs that exist in the cell can bind to repressors, leading to a reduction of the binding between repressors and promoters, leading to activation of the silent genes. They found that two TFDs, RsliM14I and RsliM14II, can activate actinorhodin, and two TFDs, RsliM18I and RsliM18III, can activate undecylprodigiosin. Further, the TFD strategy with a low-throughput screen was implicated in *S. coelicolor* A3(2), *S. albus* J1074, and *Streptomyces* sp. F-5635. This strategy was employed to activate 11 silent BGCs, whereas four silent BGCs (36% success rate) were successfully activated, in which butyrolactol A (Figure 2) was activated in *Streptomyces* sp. F-5635.

In the case of the target BGCs that synthesize color compounds, screening is easy. However, for the colorless compounds that are not easily identified (the compounds absorb in the UV region, or do not absorb the light), it increases the challenge for deducing the results of activation. To find a solution, the TFD strategy was developed by combining with reporter-guided strategy and using the high-throughput screening approach. The TFDs were constructed with a reporter gene (example: GFP) to make a library of vectors. This library of vectors was transformed into native *Streptomyces* host, and screening based on the expression of the reporter gene can assist in the selection of the recombinant strains (Figure 2). This strategy was used for activating 10 silent BGCs in *Streptomyces griseofuscus* B-5429 and *Streptomyces* sp. F-4335 responsible for biosynthesis of oxazolepoxidomycin A (Figure 2), whereas five of them (50% success rate) were successfully activated. This strategy is simpler than other strategies that use deletion or genome editing, such as the deletion of a negative regulator. However, the efficiency of this approach is not high at about 36–50% for the activation of silent BGCs [63]. Therefore, there is still room to improve this strategy by further engineering strategies.

## 6. Promoter Replacement Strategy Using CRISPR-Cas9 Technique

In 1987, Ishino et al. sequenced the characteristic clustered regularly interspaced short palindromic repeats (CRISPR) from E. coli for the first time, but their function was not fully understood [64]. Twenty years later, there was no report about these repeats until in 2007, Barrangou et al. investigated the function of CRISPR. They found that after a viral infection, bacteria integrate spacers from the phage genome to create CRISPR, combined with Cas protein, that can resist phages by spacer–phage sequence similarity [65]. The fundamental information on the biological role of CRISPR-Cas9 and detailed studies about its mechanism of action and structural aspects have been already explored [66,67,68,69,70].

Due to its biotechnological application for precise genome engineering it was successfully applied in the mammalian system in 2013 [71]. After that, there was a revolution in the use of the CRISPR system for editing the multiplex genomes in a variety of hosts across all domains of life. The CRISPR-Cas9 system was also used for the multiplex genome editing of *Streptomyces* strains. The vectors pCRISPomyces-1 and pCRISPomyces-2 derived from suicide vector pKC1139 were widely used in the genome editing of *Streptomyces*. The pCRISPomyces vectors allow the easy assembly of gRNA and editing of templates by Golden Gate assembly or the conventional digestion/ligation method. By using the CRISPR-Cas9 system, the *redN* from the BGC of undecylprodigiosin, and *actVA-ORF5* from the BGC of actinorhodin in *S. lividans* 66, *phpD* and *phpM* from the BGC of phosphinothricin tripeptide in *S. viridochromogenes* DSM 40736 were successfully deleted [72]. In addition, they also successfully deleted the full lanthipeptide BGC in *S. albus* J1074 [72]. Based on the same platform, a pKCcas9dO vector was constructed for editing the genome of *S. coelicocor* M145, including the single gene deletion, as *actII-orf4, red*, and *glnR*, and the full BGC of actinorhodin (23.3 kb), undecylprodigiosin (31.6 kb), and Ca^2+^-dependent antibiotic (82.8 kb). The multiplex gene deletion of *actII-orf4* from the actinorhodin BGC and *redD* from the undecylprodigiosin BGC was manifested by the co-expression of Cas9 and multi-sgRNA expression modules. Furthermore, the point mutation of *rpsL* was successfully induced by CRISPR-Cas9 [73]. Therefore, CRISPR-Cas9 represents a time-saving and highly efficient tool for genome editing in *Streptomyces* strains.

The promoter replacement strategy is a very efficient approach for activating silent BGCs in *Streptomyces*. The constitutive promoter as *ermE** was knocked in on the upstream region of core genes of NRPS gene cluster and PKS-NRPS gene cluster in *S. albus* J1074, resulting in activation of the blue indigoidine (Figure 3B) and polycyclic tetramate macrolactam family as 6-epi-alteramides A and B (Figure 3B) [74]. Later, the promoter replacement strategy was developed by applying CRISPR-Cas9 to activate the silent gene clusters in streptomycetes. KasOp* promoter was used for the knock-in experiments to native hosts (Figure 3A) [75,76]. Ten silent biosynthesis gene clusters in five *Streptomyces* strains, namely *S. albus*, *S. lividans*, *Streptomyces roseosporus* NRRL15998, *Streptomyces venezuelae* ATCC10712, and *S. viridochromogenes* DSM 40736, were activated by replacing native promoter by KasOp* promoter of the core genes of silent gene clusters. These biosynthesis gene clusters were involved in types I, II, and III PKSs, non-ribosomal peptide synthetase (NRPS), hybrid PKS–NRPS, and phosphonate clusters. The editing efficiency per total screened strains was very high from 50% to 100%, and the efficiency of activation was 100%, whereas a novel pentangular type II polyketide (Figure 3B) was discovered in *S. viridochromogenes* and three different types of compounds as macrolactam, photocyclized alteramide A and FR-90098 (Figure 3B) were produced by engineered *S. roseosporus* [76]. More recently, the production of auroramycin (Figure 3B), a type I PKS, was successfully activated in *S. roseosporus* NRRL 15998 by using CRISPR-Cas9 based knock-in of KasOp* promoter its silent type I PKS BGC [77].

The promoter replacement is the very efficient strategy for the activation of the cryptic BGCs. The only limitation of this strategy is that in some strains, the genetic manipulation is not amenable [77]. Overall, this strategy has the potential to become a key method to activate cryptic biosynthesis gene cluster in *Streptomycetes*, because of the high efficiency in genome editing, versatility in application in diverse *Streptomyces* strains or most types of BGCs, and being easy to apply, and saving time. 

## 7. Recent Approaches for Discovery/Characterization of Cryptic Natural Products

In addition to the systematic activation of cryptic BGCs, there is equivocal requirement for parallel developments in techniques for bioactivity screening, isolation and separation methods, and analytical chemistry [19,78]. Recently, there has been an enormous development in robust analytical techniques and comparison with reference databases in both genomic and metabolite profile levels. Basically, the approaches have been categorized as top-down approaches based on metabolomics studies and bottom-up approaches centered on genomics studies and finally interconnecting both of them [78].

### 7.1. Top-Down Approach Based on Metabolomics

The top-down approach is based on prior analysis of metabolites and subsequent analysis of the genomic data. Metabolomics is an effective analytical technique that encompasses the profiling of low molecular weight metabolites between medium and high throughput environments. In this approach, different biological samples are statistically analyzed and correlated, with the activity of interest focusing on differentially produced compounds as potential biomarkers. Therefore, metabolomics has the potential to lead to the discovery of novel NPs [79,80]. Recently, the development in analytical techniques, such as nuclear magnetic resonance (NMR) spectroscopy and mass spectrometry analysis has advanced the metabolomics approach to a new horizon. Different NMR techniques, such as J-resolved, COSY, TOCSY, HSQC, and HMBC, have the potential to precisely elucidate the chemical structure of the natural product. However, it is ineffective on impure organic compounds, because there are chances of signal-overlapping issues. To address these types of complexities, chromatography-hyphenated NMR (HPLC-NMR) has been introduced into metabolic analysis [81]. The incorporation of solid-phase-extraction (SPE) for injection sample refinement or integration of mass spectrometric instrumentation can fine-tune the structural evaluation of chemical components in NMR-based metabolomics studies. By using this LC-SPE-NMR approach [82] and LC-DAD-SPE-MSNMR instrumentation [83] comprehensive structural information can be deduced. Hence, it is feasible to get insight into the chemical composition of crude extracts through in-silico analysis leading to the assessment of the valuable components present on the extract or fraction at the early stage. 

A workflow of NMR-based metabolomics and bioinformatics was utilized for the identification of novel pyranonaphtoquinones. The constitutive expression of the pathway-specific activator, *qin* in *Streptomyces* sp. MBT76 resulted in the activation of the biosynthesis of a cryptic type II PKS compound belonging to the family of pyranonaphtoquinones. The genomics analysis facilitated by structural evaluation identified the chemical structures belonging to the family of 8-C-glycosyl-pyranonaphthoquinones. The qinimycins A–C possessed structural features including a rare 5,14-epoxidation and unprecedented 13-hydroxylation in pyranonaphthoquinones. A deoxyaminosugar was unusually attached to the pyranonaphthoquinone backbone similar to BE-54238A and BE-54238B [84].

The continued advances in the large-scale acquisition and analysis of metabolomic data has made it possible to thoroughly process complex metabolite samples leading to the discovery of novel small molecules. The metabolomics-based comparison of supernatants of *Streptomyces chartreusis* cultivated in different media, using liquid chromatography-coupled with tandem MS, and further analysis by molecular networking, led to the characterization of diverse NPs, including the isolation of a new polyether ionophore, named deoxacalcimycin [85]. By employing genome mining the previously unreported capability of a marine sponge-derived isolate, namely *Streptomyces* sp. SM17, to produce surugamide A was identified [86]. Similarly, the detailed metabolomic analysis of 4160 fractions obtained from 13 actinomycetes maintained under 32 different culture conditions was performed by ^1^H NMR spectroscopy and multivariate analysis [87]. Metabolomics screening using liquid chromatography-high resolution mass spectrometry enabled the identification and purification of terrosamycin A and B, and two polycyclic polyethers that are active against breast cancer from the fermentation broth of *Streptomyces* sp. RKND004 [88].

### 7.2. Bottom-Up Approaches Based on Genomics

The bottom-up approach based on genomics has been utilized to unveil new NPs that are usually undetected under standard fermentation conditions [89]. These strategies incorporate the combination of functional genomics and bioinformatics to identify the products of activated BGCs. The recent advances in genome sequencing have not only eased the robust availability of genomic information of *Streptomyces*, but also the availability of high-quality data can be utilized by different bioinformatic tools. The interrogation of genome information by these bioinformatics tools provides the effective approach for connecting particular BGCs to specific NPs. The diverse bioinformatics tools such as ClustScan [90], antiSMASH 5.0 [91], ClusterFinder [92], PRISM [93], ARTS [94], BAGEL4 [95], RiPPER [96], and RODEO [97] are utilized for analysis of diverse BGCs responsible for formation of PKS, NRPS, and RiPP compounds. Thus, concrete information about BGCs can assist in designing systematic isolation techniques and structural elucidation of NPs based on correlation of genomic information with the metabolite profile. Moreover, the availability of a global repository of BGCs from different microbes has also assisted with the precise prediction of the BGCs based on comparative analysis of BGCs in databases [98]. More recently, the advances in the bioinformatic exploration of genomic data have enabled the sophisticated computational framework for systematic exploration of BGCs [99].

## 8. Conclusions and Future Perspectives

In this review, we summarized several versatile methods that are used to activate silent natural product BGCs by the manipulation of regulatory genes, ribosomal engineering, co-cultivation, or precise genome engineering mediated by CRISPR-Cas9. However, there are advantages and disadvantages to each method. For example, although the particular biosynthetic pathway-specific manipulation can link the product and the gene cluster directly, it can only activate one specific gene cluster at a time. The manipulation of the global regulatory genes, ribosomal engineering, and cultivation in varied conditions may activate several BGCs at a time, but it is difficult to link the specific metabolites with the BGCs. For those methods toward multiple BGC activation, if an easy way can be found to link the particular BGCs to their specific products, respectively, then the high-throughput activation of cryptic biosynthetic pathways can become feasible. In other instances, the revolutionary technology for precise genome engineering like CRISPR-Cas9 has disadvantages, such as the non-specificity of DSBs, the toxicity of Cas nuclease, and poor expression levels of *cas9* gene or *sgRNAs*, which have been major bottlenecks for their application in multiplexed fashion [100]. However, there have been attempts to improvise the application of CRISPR-Cas9 by employing conjugating with base-editing enzymes [101] and in the future, many breakthroughs can update the application of these approaches. The application of CRISPR interference (CRISPRi) [102] and CRISPR activation (CRISPRa) [103] in addition to gene deletions in multiplexed fashion, can bring forth rational metabolic engineering approaches for activating cryptic BGCs mediated by multi-repurposing of precursor pathways, regulators, and biosynthetic enzymes. 

The ease of availability of the genome sequence of diverse *Streptomyces* has unfolded the genomic context of all the BGCs, including cryptic BGCs, that can be analyzed by versatile in silico based BGC prediction software (ActinoBase) [104]. Nevertheless, these tools provide an effective prediction of diverse NPs belonging to different compound classes as PKS, NRPS, RiPPs, or hybrids, but still much greater robustness and higher precision can be expected. In addition, the availability of web-based tools, such as MASST [105], have enabled high-throughput screening of NPs based on their mass fragmentation patterns, which can enable the precise prediction of the metabolite from the culture broth, independent of the purification and structural elucidation steps. Furthermore, the available knowledge resources from genomics and metabolomics can be employed for strain engineering, biosynthetic pathway engineering, synthetic biology, systems biology, and media optimization technology for the production of such cryptic/known biomolecules in significant titers [106]. Such burgeoning development in both genetic studies, as well as metabolic profiling, can be further harnessed in metabolic engineering strategies by the application of computational approaches, such as artificial intelligence (AI). These approaches can be effective in simulating the connection between the genomics and metabolomics to generate intelligence in these production hosts, so that they can sense the environmental condition, and respond rationally [107]. Therefore, the next dimension for the activation of cryptic BGCs can be based on computational optimization of genomics, proteomics, and metabolomics, based on machine learning and AI. Such information can be utilized for the reincarnation of a particular production host by robust synthetic biological tools, or genome engineering approaches based on the predicted mathematical models and simulations. 

## Figures and Tables

**Figure 1 microorganisms-08-00616-f001:**
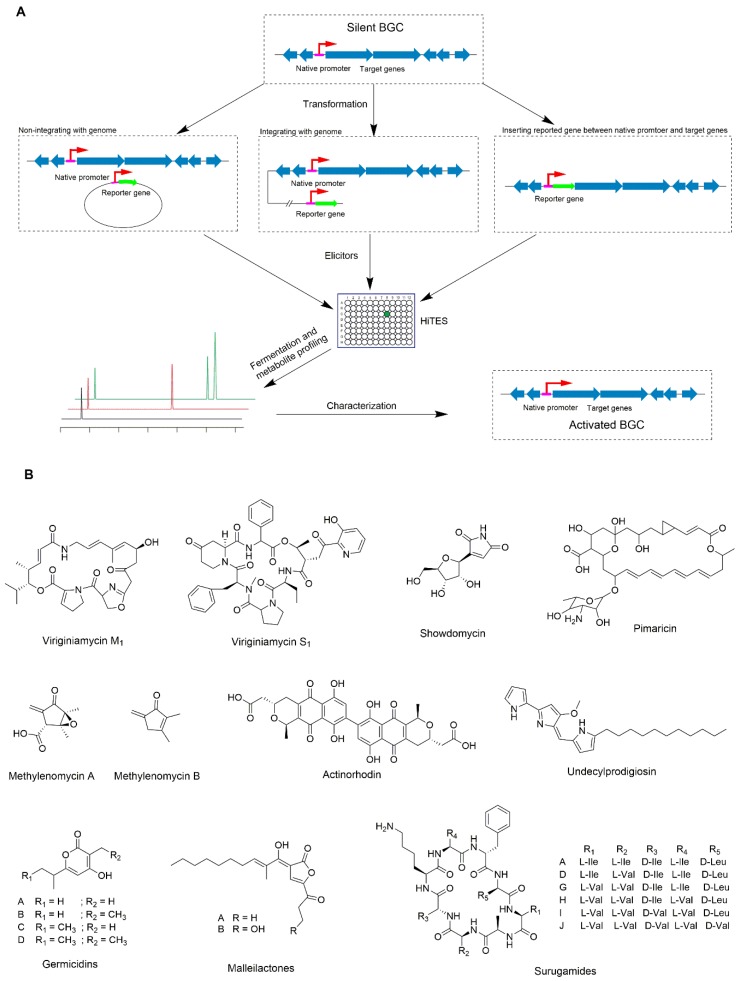
Overview of High-Throughput Elicitor Screens (HiTES) combined reporter-guided screening approach using rational elicitor. (**A**) Overview of HiTES combined reporter-guided screening approach using rational elicitor. The red arrows represent the promoter. The green arrows represent the reporter gene. (**B**) Structure of compounds are activated by HiTES combined reporter-guided screening approach using rational elicitor.

**Figure 2 microorganisms-08-00616-f002:**
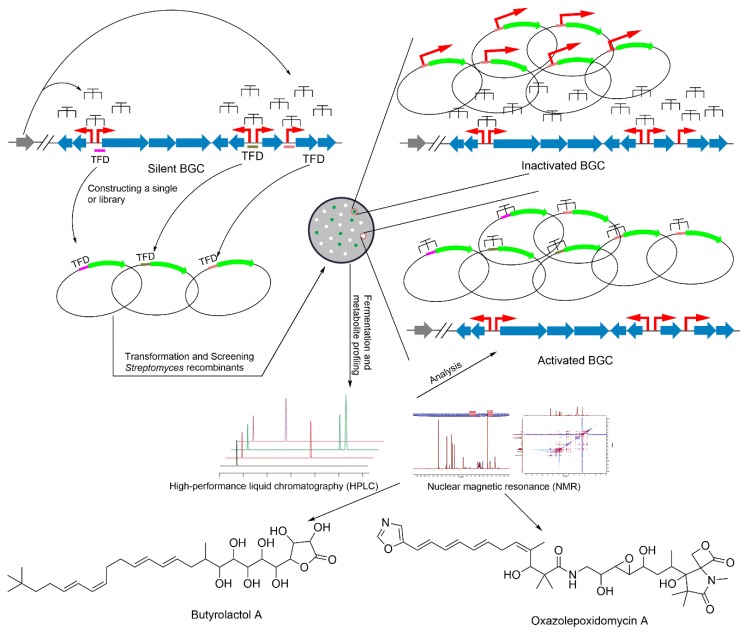
Overview of transcription factor decoy (TFD) strategy. The red arrows represent the promoter. The green arrows represent the reporter gene.

**Figure 3 microorganisms-08-00616-f003:**
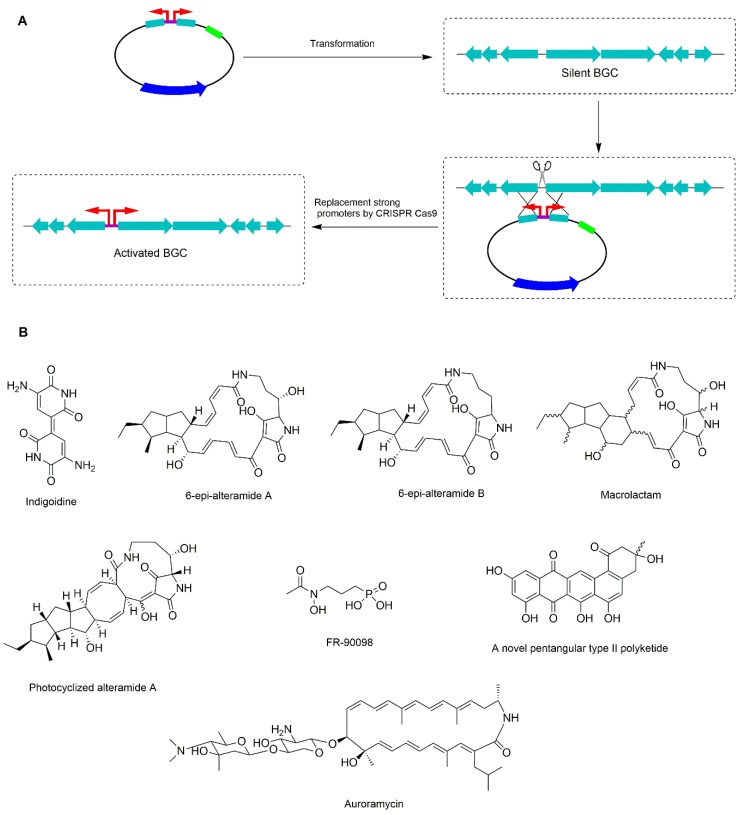
Overview of promoter replacement strategy using CRISPR-Cas9 technique. (**A**) Promoter replacement strategy using CRISPR-Cas9 technique. The dark blue arrows represent for Cas9 gene. The green arrows represent for gRNA. The light blue arrows represent for homology arms. (**B**) Structure of compounds are activated by promoter replacement strategy.

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
