# Peer review of "Recent Advances in Strategies for Activation and Discovery/Characterization of Cryptic Biosynthetic Gene Clusters in Streptomyces"

_microorganisms, 2020, doi:10.3390/microorganisms8040616_

Round 1

Reviewer 1 Report

The review provides a detailed overview of the state of the art in research and method development for activation of cryptic clusters.

A couple of suggestions to improve the manuscript:

General comments:

1- I find the manuscript hard to read with sentences or part of sentences in many places that are not easy to understand or require guesswork to understand what the authors try to say. Therefore, I suggest to have a native English speaker taking a good look at the manuscript.

2- The many figures with compound structures give the impression that the focus is on the compounds produced rather than on the methods currently developed to wake up cryptic clusters. For the general reader replacing some of the five figures with structures with figure(s) illustrating for example the complexity and size of the cryptic clusters. Even more because the structures in these figures are not really discussed in the text.

3- What is missing and should be added is a section with a thorough overview of how BGS’s are linked by computational and genetic analysis to a given product.

Line 29-30:  The biological role of these compounds might be quite different from the biological use that humans make of these compounds. In the context of waking up cryptic BGC’s it could be of interest to pay attention to views like discussed recently by Pishchancy & Kolter (doi.org/10.1111/mmi.14471) on the possible ecological role of antimicrobials.

Line 78-80: Not clear what the authors wish to state here.

Line 100: It should be made clear what these numbers in boldface refer to.

Line 114-119: 2 out of 7 Streptomyces strains, 5 out of 61 non-streptomyces strains, and so on.

Line 173: Figure 2C, I don’t see the co-cultivation in this drawing

Line 246: They integrated the reported gene in the genome (?) downstream of Psur

Line 206-256: Good to add a discussion on whether these chemical elicitors have any relation to known small molecule metabolites. This will give a better view on whether these chemical elicitors can teach us something on how the cryptic clusters are activated in the environment. Knowledge of the natural elicitors would be a major step forward in this research field.

Line 296: “The immune systems of prokaryotes” what does it refer to?

Line 295-328: No need to give these details of the CRISPR Cas system, better to refer to a recent review on Crispr and start this section with line 329.

Line 328: Replace “23 strains showed new peaks” by 23 strains showed new products

Line 446: “orthogonal basis”, not clear what this means in the context of the sentence.

Line 448-449: that each method has merits and demerits has already ben stated in line 440, so it can be deleted here.

Author Response

  1. Some figures of compound structures were removed as requested.
  2. Line 29-30: Revised as requested.

Line 78-80: Revised as requested.

Line 100: Revised as requested (line 102 in corrected manuscript).

Line 114-119: Revised as requested (line 124-124 in corrected manuscript).

Line 173: Figure is removed.

Line 246: (line 260 in corrected manuscript) The figure is show for principle of HiTES combined reported-guided screening approach. It is not only for activating the sur gene cluster, therefore this approach can be done by inserting the reporter gene between native promoter and target gene.

Line 296: (line 326 in corrected manuscript) Revised as requested.

Line 295-328: (line 324-361 in corrected manuscript) Revised as requested.

Line 446: (line 513 in corrected manuscript) Revised as requested.

Line 448-449: (line 514-516 in corrected manuscript) Revised as requested.

Reviewer 2 Report

Title: Recent advance strategies for activating cryptic biosynthesis gene cluster in Streptomyces

Paper Summary: The authors aim to summarize recent strategies to activate cryptic BGCs in Streptomyces. They begin by highlighting the historical impact of Streptomyces in natural product drug discovery/development efforts. The idea of a cryptic BGC is introduced, and the need to develop new strategies to identify compounds encoded by these cryptic BGCs is justified.

                From there, the authors summarize recent (and at times not so recent) examples of NP discovery projects that fall into seven categories: (1) Ribosome/RNAP engineering, (2) OSMAC, (3) HiTES, (4) Transcription factor decoys, (5) Refactoring, (6) Protein post-translational modification with PPTase, and (7) Analytical Chemistry approaches [section titles changed/paraphrased by me]. For each section, the authors briefly describe the approach and cite many specific examples. Figures at times illustrate the approaches being employed, and at times merely show structures of molecules produced via different techniques.

Summary of this critical review:

The largest weakness in this review is that approaches to genome mining natural products in Streptomyces have been reviewed by many others already, and this paper does not seem to contribute anything substantially new to that body of literature. There are at least two ways I can illustrate this objectively. (i) Searching PubMed with the key words “genome mining of natural products review” retrieves 118 papers from the last 5 years alone. (ii) if I search for papers that cite the same techniques/approaches that are reviewed herein, I find that the Transcription Factor Decoy approach is cited in 6 reviews, the 2017 HiTES paper is cited in 14 reviews, etc. It almost seems like there are more reviews of the recent advances in NP discovery than there are new advances. The problem with this is that it will be difficult for the current review to stick out from the rest (which it would need to do to become well-cited). I think the authors would be better served by picking a unique/novel perspective and writing from there.

                I suggest that the authors spend more effort articulating why this review constitutes a new and needed contribution to the current literature. Other weaknesses are listed below.

  1. Grammar and language usage. There are several grammar/language problems with this manuscript that limit its readability. Normally when I review I include a point-by-point list of grammar edits/revisions, but there are simply too many to provide such a list here.
  2. The authors state that “in recent years, the chances of discovery of novel bioactive compounds from Streptomyces have significantly declined”. What evidence do they have for this? I note that Pye et Linington (https://doi.org/10.1073/pnas.1614680114) find that new NP discovery rates have not been in decline, despite the common claim that they have been.
  3. In Figure 2, do you have permission to use that photo? You should at least cite the source of that photo, and the editor should probably check to make sure that appropriate permissions have been secured.
  4. Why include the traditional OSMAC approach in a review titled ‘Recent advances…’
  5. Figure 6 is unclear to me. Each colony is transformed with the same plasmids, but sometimes the reporter is expressed and sometimes it is not?
  6. The term ‘refactoring’ is misused throughout the manuscript. Unfortunately in recent years, refactoring has been increasingly used for ‘promoter replacement’. This is far more limited in scope than the original use of the term in the synthetic biology literature (Chen, Kosuri, & Endy 2005 Mol Sys Biol). I recommend using more precise language such as ‘promoter replacement’ when appropriate throughout the manuscript. As it is written, ‘refactoring’ is just jargon that is used incorrectly.
  7. As with the OSMAC section, protein post-translational modification is not really a recent advance. It was recently applied in high throughput, but it seems out of place in this review (i.e. should not be described on the same level as HiTES, Transcription Factor Decoys, CRISPR-Cas9 genome editing, etc.)
  8. The mechanism by which PPTase expression works is not likely to be correct. I do not believe there is any direct evidence that the endogenous PPTase has been mutated, as these strains all produce fatty acids and other PKs, suggesting there are still functional PPTases encoded in the genome.
  9. Section 8 (NMR and MS approaches) also seems out of place, as these are not methods to awaken cryptic BGCs (i.e. the title of the paper).

Author Response

  1. We use the reference (https://doi.org/10.1073/pnas.1614680114) to revise as requested (line 51-61 in corrected manuscript)
  2. Figure 2 is removed, Revised as requested.
  3. Traditional OSMAC is replaced by “The OSMAC approach mediated by co-culture”
  4. In Figure 6 (Figure 5 in corrected manuscript), in high-throughput screening, a library of TFD-contained vectors is constructed. Then, transforming this library to native Streptomyces. The transformant colonies have different TFD-contained vectors, therefore, the reporter gene can be expressed or not expressed, as the figure is showed.
  5. The term “refactoring” are replaced by “promoter replacement” as requested.
  6. The protein post-translational modification section is removed ad requested.
  7. The Section 8 (NMR and MS approaches) are changed to section 7 (Recent approaches for discovery/characterization of cryptic natural products)

Round 2

Reviewer 2 Report

Summary

The revised manuscript is much improved over the original submission. Unfortunately, my #1 criticism (grammar and language usage) was not addressed. As before, there are simply too many errors for me to provide point-by-point corrections. I included some corrections (up to line 68) below.

Notes on revision: (blue text from my original review)

  1. Grammar and language usage. There are several grammar/language problems with this manuscript that limit its readability. Normally when I review I include a point-by-point list of grammar edits/revisions, but there are simply too many to provide such a list here.
    1. This criticism was not addressed in the revision. I highlight several grammatical errors in the first 68 lines below.
  2. The authors state that “in recent years, the chances of discovery of novel bioactive compounds from Streptomyces have significantly declined”. What evidence do they have for this? I note that Pye et Linington (https://doi.org/10.1073/pnas.1614680114) find that new NP discovery rates have not been in decline, despite the common claim that they have been.
    1. The authors revised the review accordingly.
  3. In Figure 2, do you have permission to use that photo? You should at least cite the source of that photo, and the editor should probably check to make sure that appropriate permissions have been secured.
    1. The figure in question has been removed.
  4. Why include the traditional OSMAC approach in a review titled ‘Recent advances…’
    1. The authors have revised this to highlight co-culturing
  5. Figure 6 is unclear to me. Each colony is transformed with the same plasmids, but sometimes the reporter is expressed and sometimes it is not?
    1. The figure is more clear, but could still use a legend. For example, it appears that the length of the red promoter icon is meant to correspond to gene expression level. If this is true, it should be explicitly stated in the figure legend. Also, it is not obvious from the drawing that the plasmids on the left are meant to represent a diverse library, since they are all drawn identically.
  6. The term ‘refactoring’ is misused throughout the manuscript. Unfortunately in recent years, refactoring has been increasingly used for ‘promoter replacement’. This is far more limited in scope than the original use of the term in the synthetic biology literature (Chen, Kosuri, & Endy 2005 Mol Sys Biol). I recommend using more precise language such as ‘promoter replacement’ when appropriate throughout the manuscript. As it is written, ‘refactoring’ is just jargon that is used incorrectly.
    1. Adequately corrected in the revision.
  7. As with the OSMAC section, protein post-translational modification is not really a recent advance. It was recently applied in high throughput, but it seems out of place in this review (i.e. should not be described on the same level as HiTES, Transcription Factor Decoys, CRISPR-Cas9 genome editing, etc.)
    1. PTM is no longer described as a recent strategy.
  8. The mechanism by which PPTase expression works is not likely to be correct. I do not believe there is any direct evidence that the endogenous PPTase has been mutated, as these strains all produce fatty acids and other PKs, suggesting there are still functional PPTases encoded in the genome.
    1. This section has been deleted.
  9. Section 8 (NMR and MS approaches) also seems out of place, as these are not methods to awaken cryptic BGCs (i.e. the title of the paper).
    1. This section fits into the overall context of the manuscript much better, with the introductory paragraph and the additional section on bottom-up approaches.

Sample grammatical errors:

L2: “Recent advances in strategies…”

L3: “cryptic biosynthetic gene clusters in …”

L14: “prolific sources”

L15: “anticancer agents” or “anticancer chemotherapeutics”

L15: “immunosuppressants”

L20: ‘cryptic’ is not typically used to describe compounds, but biosynthetic gene clusters. The usage here is incorrect. Recommended change: “… significant titers to support drug discovery/development efforts”. (the usage of ‘cryptic’ in L26 is correct)

L22: “their entire network of metabolism and regulatory system” is grammatically incorrect, but I am not sure what you are trying to say, so cannot offer a correction.

L41: “are able to produce”

L44: “Streptomyces is a genus of Gram-positive…”

L47: “colleagues were pioneers for”

L51: “indicated that the rate of new NP discovery has been maintained despite a drop in the number of compounds making it through regulatory approval pipelines”

L53: ‘natural products’ should be abbreviated ‘NPs’. Improper use of ‘cryptic’ to describe a compound, instead of a BGC.

L58: “In addition, advances in the studies…”

L59: “illustrate the entire”

L60: missing space between sentences

L61: what is meant by “landmarked proper connection”?

L63: “a larger diversity of NPs than those isolated to date”

L68: rewrite: “for manifesting the production of cryptic compounds”, for example as, “for obtaining isolable quantities of compounds usually produced in low titers”

Author Response

Respected Editor-in-Chief,

Microorganisms

We are thankful to editors and reviewers for their critical comments and suggestions for revising the manuscript. We have duly revised the manuscript considering the comments. The responses to the comments have been highlighted in blue color. However, the entire manuscript has been thoroughly revised regarding and we hope that the revised version is suitable for wider readership.

Sincerely,

Prof. Jae Kyung Sohng

Major Comments:

Unfortunately, my #1 criticism (grammar and language usage) was not addressed. As before, there are simply too many errors for me to provide point-by-point corrections. I included some corrections (up to line 68) below.

Response: We have thoroughly revised the manuscript considering the grammar and language use. We have also consulted with www.harrisco.com

Sample grammatical errors:

Comments 1: L2: “Recent advances in strategies…”

Response: The correction has been as shown in blue color.

Comment 2: L3: “cryptic biosynthetic gene clusters in …”

Response: The correction has been as shown in blue color.

Comment 3: L14: “prolific sources”

Response: The correction has been as shown in blue color.

Comment 4: L15: “anticancer agents” or “anticancer chemotherapeutics”

Response: The correction has been as shown in blue color.

Comment 5: L15: “immunosuppressants”

Response: The correction has been as shown in blue color.

Comment 6: L20: ‘cryptic’ is not typically used to describe compounds, but biosynthetic gene clusters. The usage here is incorrect. Recommended change: “… significant titers to support drug discovery/development efforts”. (the usage of ‘cryptic’ in L26 is correct)

Response: The correction has been as : the biosynthetic gene clusters (BGCs) are cryptic.”

Comment 7: L22: “their entire network of metabolism and regulatory system” is grammatically incorrect, but I am not sure what you are trying to say, so cannot offer a correction.

Response: The correction has been made as the studies of metabolomics provide a portfolio of entire metabolites produced from the strain of interest

Comment 8: L41: “are able to produce”

Response: The correction has been as shown in blue color.

Comment 9: L44: “Streptomyces is a genus of Gram-positive…”

Response: The correction has been as shown in blue color.

Comment 10: L47: “colleagues were pioneers for”

Response: The correction has been as shown in blue color.

Comment 11: L51: “indicated that the rate of new NP discovery has been maintained despite a drop in the number of compounds making it through regulatory approval pipelines”

Response: The correction has been made as that the rate of new NP discovery has been maintained despite a drop in the number of compounds making it through regulatory approval pipelines,

Comment 12: L53: ‘natural products’ should be abbreviated ‘NPs’. Improper use of ‘cryptic’ to describe a compound, instead of a BGC.

Response: The correction has been as shown in blue color.

Comment 13: L58: “In addition, advances in the studies…”

Response: The correction has been as shown in blue color.

Comment 14: L59: “illustrate the entire”

Response: The correction has been as shown in blue color.

Comment 16: L60: missing space between sentences

Response: The correction has been as shown in blue color.

Comment 17: L61: what is meant by “landmarked proper connection”?

Response: The correction has been as have established

Comment 18: L63: “a larger diversity of NPs than those isolated to date”

Response: The correction has been as shown in blue color.

Comment 19: L68: rewrite: “for manifesting the production of cryptic compounds”, for example as, “for obtaining isolable quantities of compounds usually produced in low titers”

Response: The correction has been made as have been successfully utilized for obtaining isolatable quantities of compounds usually produced in low titers,